# New Genes Identified as Modulating Salt Tolerance in Maize Seedlings Using the Combination of Transcriptome Analysis and BSA

**DOI:** 10.3390/plants12061331

**Published:** 2023-03-15

**Authors:** Yongxing Zhu, Ying Ren, Ji’an Liu, Wenguang Liang, Yuanyuan Zhang, Fengyuan Shen, Jiang Ling, Chunyi Zhang

**Affiliations:** 1Biotechnology Research Institute, Chinese Academy of Agricultural Science, Beijing 100081, China; 2Agricultural Biotechnology Center, Ningxia Academy of Agriculture and Forestry Sciences, Yinchuan 750002, China; 3Sanya Institute, Hainan Academy of Agricultural Sciences, Sanya 572000, China

**Keywords:** salt-tolerant, BSA-seq, transcriptomic analysis, maize inbred

## Abstract

(1) Background: Salt stress is an abiotic factor that limits maize yield and quality. A highly salt-tolerance inbred AS5 and a salt-sensitive inbred NX420 collected from Ningxia Province, China, were used to identify new genes for modulating salt resistance in maize. (2) Methods: To understand the different molecular bases of salt tolerance in AS5 and NX420, we performed BSA-seq using an F2 population for two extreme bulks derived from the cross between AS5 and NX420. Transcriptomic analysis was also conducted for AS5 and NX420 at the seedling stage after treatment with 150 mM of NaCl for 14 days. (3) Results: AS5 had a higher biomass and lower Na^+^ content than NX420 in the seedling stage after treatment with 150 mM NaCl for 14 days. One hundred and six candidate regions for salt tolerance were mapped on all of the chromosomes through BSA-seq using F2 in an extreme population. Based on the polymorphisms identified between both parents, we detected 77 genes. A large number of differentially expressed genes (DEGs) at the seedling stage under salt stress between these two inbred lines were detected using transcriptome sequencing. GO analysis indicated that 925 and 686 genes were significantly enriched in the integral component of the membrane of AS5 and NX420, respectively. Among these results, two and four DEGs were identified as overlapping in these two inbred lines using BSA-seq and transcriptomic analysis, respectively. Two genes (*Zm00001d053925* and *Zm00001d037181*) were detected in both AS5 and NX420; the transcription level of *Zm00001d053925* was induced to be significantly higher in AS5 than in NX420 (41.99 times versus 6.06 times after 150 mM of NaCl treatment for 48 h), while the expression of *Zm00001d037181* showed no significant difference upon salt treatment in both lines. The functional annotation of the new candidate genes showed that it was an unknown function protein. (4) Conclusions: *Zm00001d053925* is a new functional gene responding to salt stress in the seedling stage, which provides an important genetic resource for salt-tolerant maize breeding.

## 1. Introduction

As a major environmental abiotic stress, soil salinity affects more than 7% of the world’s land area, and the total area of saline soil in China accounts for 4.88% of the country’s total available land [1,2]. Large, negative impacts have been observed on the yields of crops such as wheat (*Triticum aestivum* L.), maize (*Zea mays* L.), rice (*Oryza sativa* L.), and cotton (*Anemone vitifolia* Buch), and cultivating salt-tolerant varieties can effectively improve these negative effects [3]. The salt tolerance mechanisms of plants have been investigated for several decades, and a range of adaptive strategies of plants for salt responses have been reported, such as hormone signaling pathways [4], transcriptional and post-transcriptional regulation [5,6,7], chromatin modification [8,9], and the salt overly sensitive (SOS) pathway.

Among studies on osmotic, ionic, and oxidative stress under saline conditions, sodium chloride (NaCl) is the most studied worldwide due to its high solubility and ubiquitous distribution [2,10]. The excessive accumulation of sodium (Na^+^) evokes osmotic damage and ionic stress, and the uptake, translocation, and compartment of Na^+^ by the SOS pathway at cellular, tissue, and whole-plant levels are essential for plant salt tolerance [11]. The removal of Na^+^ from the cytoplasm by transporting Na^+^ into the vacuole or out of the cell is conducted by SOS1, which is a plasma-membrane-localized NHX family Na^+^/H^+^ antiporter, and it is activated by the direct phosphorylation of SOS2 (a CBL-interacting protein kinase (CIPK) family kinase) [12,13]. Another way that the decrease in the cytosolic Na^+^ concentration is regulated is through ion transporters, especially those with a preference for Na^+^ or potassium (K^+^) ions, such as the Na+-selective transporters encoded by the Na^+^/H^+^ exchanger (NHX), high-affinity K^+^ transporter 1 (HKT1) family genes, and the Na^+^-selective transporter encoded by high-affinity K^+^ (HAK) genes [11,14,15,16]. As the tolerance to salinity is a kind of quantitative genetic trait, previous studies have covered only partial knowledge about plants’ adaptability.

To broaden natural genetic diversities and to acquire more valuable sources for breeding salt-tolerant crops, dozens of quantitative genetic trait (QTL) genes and candidate genes from genome-wide association studies (GWAS) have been identified to be associated with natural variations in salt tolerance [3,16,17,18,19,20]. Several other key metabolisms were found to be associated with the Na^+^ content in cells. For example, in a study, it was found that the wheat ancestral Na^+^ transporter gene NAX2 (Na^+^ Exclusion 2) could increase the yield by 25% under salinity [21]. In other studies, it was found that an elite allelic variation of rice SKC1 (Shoot K^+^ Content 1) promotes salt tolerance by regulating K^+^/Na^+^ homeostasis [22], the maize chloride transport regulation of ZmCLCg could increase plants’ tolerance to salinity [23], and shoot Na^+^ variations were conferred by ZmNSA1 (Na^+^ content under saline–alkaline conditions) under NaHCO_3_ conditions [18]. This research indicates that natural genetic diversities provide valuable sources for identifying new candidate genes for breeding salt-tolerant crops. Recently, new strategies that combine several emerging technologies (Bulked Segregate Analysis, RNA-sequencing, metabolome analysis, or phosphoproteomic analysis) have been carried out to find new genes associated with several traits, such as the anthocyanin in cucumber (*Cucumis sativus* L.) fruit skin [24], the panicle grain number in rice [25], and the plant architecture in *Brassica napus* [26].

In this study, BSA-seq and RNA-seq were combined to find new candidate maize genes involved in leaf tolerance to salinity using two inbred lines, named “NX420” and “AS5”, which show differential tolerance levels to salinity. Two candidate genes were screened using the combination of RNA-seq from NaCl-treated inbred seedlings for 14 days and BSA-seq. *Zm00001d053925* showed an inducible response at the transcriptional level upon NaCl treatment in NX420 and AS5, which were collected from Ningxia Province, China. Protein structure prediction indicated that *Zm00001d053925* contained five transmembrane domains. These results indicate that *Zm00001d053925* could be a new gene for salinity tolerance, with an unknown function.

## 2. Results

### 2.1. Detection of the Salinity Tolerance of AS5 and NX420 and Construction of the Extreme Mapping Population for BSA-seq

In this study, AS5 and NX420, collected from Ningxia Agricultural Institute, Ningxia Province, China, were used to investigate the genetic basis underlying salt tolerance in maize. We compared the phenotypes upon salt treatment between AS5 and NX420 under 150 mM of NaCl for 14 days. A moderate level of salt (150 mM NaCl) caused a significant reduction in plant height and biomass at the seedling stage (Figure 1A–C). AS5 showed a salt tolerance phenotype of a higher plant height and a lower loss of biomass, while NX420 already had dry heart leaves and showed a salt hypersensitive phenotype of shorter leaves and a lower biomass (Figure 1A–C). The salt-grown maize seedlings showed an increased leaf Na^+^ content and decreased leaf K^+^ content, which in turn caused an ion imbalance (i.e., reduced K^+^:Na^+^ ratios) [21,22,27]. The Na^+^ content was significantly lower in AS5 than that in NX420 leaves, and no differences were observed in the roots (Figure 1D); the K^+^ content was significantly higher in AS5 leaves (Figure 1E), which led to the K^+^: Na^+^ ratios being significantly higher in AS5 than those in NX420 leaves (Figure 1H). These results indicate that AS5 was more salt-tolerant than NX420.

To map the genes related to salinity tolerance in AS5, we constructed an F_2_ population derived from a cross between the salt-tolerant inbred AS5 and the salt-sensitive inbred NX420. This F_2_ population was used to identify the degree of salinity tolerance in the seedling stage in 150 mM of NaCl treatment for 14 days. Over 500 F_2_ seeds were analyzed, and 30 lines were screened to show salt tolerance (such as AS5 and LA1-30) and salt sensitivity (such as NX420 and LS1-30), using plant height to differentiate between the salt-tolerant lines and the salt-sensitive lines. Under 150 mM of NaCl treatment, the plant height survey and analysis of the two extreme populations showed that the plant height difference between the two extreme populations was significant, which was suitable for follow-up research (Figure 1I). So these lines were used to construct the tolerant and sensitive salinity bulks for BSA-seq.

### 2.2. Sequencing and Mapping of Reads to the B73 Reference Genome

Whole genome re-sequencing data were generated for both AS5 and NX420, the extreme salinity-tolerant bulk (tolerant, LA1-30), and the salinity-sensitive bulk (sensitive, LS1-30) (Table 1). A total of 61.67 million clean reads were generated for the salinity-tolerant parent, AS5; 62.08 million clean reads were generated for the salinity-sensitive parent, NX420; 78.85 million clean reads were generated for the tolerant bulk; and 75.73 million clean reads were generated for the sensitive bulk. The sequencing depths were 43.33× for AS5, 43.61× for NX420, 55.39× for the tolerant bulk, and 53.21× for the sensitive bulk. The properly paired ratios were more than 93% (93.14 for AS5, 95.29% for NX420, 94.25 for the tolerant bulk, and 94.41% for the sensitive bulk).

### 2.3. Candidate Genes for Salinity Tolerance by BSA Based on Genomic Resequencing 

In total, 6,516,313 SNPs were detected between the two bulks. There were a total of 657,086 SNPs on Chromosome 1, 639,709 SNPs on Chromosome 2, 878,652 SNPs on Chromosome 3, 668,120 SNPs on Chromosome 4, 642,954 SNPs on Chromosome 5, 598,968 SNPs on Chromosome 6, 690,312 SNPs on Chromosome 7, 625,423 SNPs on Chromosome 8, 598,375 SNPs on Chromosome 9, and 516,732 SNPs on Chromosome 10 (Table 2). An association analysis of salinity tolerance and polymorphic markers was performed using the Euclidean distance (ED) and SNP/InDel-index methods. One hundred SNPs were identified after the results were obtained based on these two methods being overlapped (Appendix A) and based on the threshold value of the confidence interval of Δ(SNP/InDel-index) at the 99% significance level (top 1%, blue line) (Figure 2A) and the ED at the 99% significance level (top 1%, pink line) (Figure 2B).

### 2.4. Transcriptome Analysis of Two Parents at the Seedling Stage under Salt Stress 

To understand the differentially expressed genes responding to salt stress, we also performed transcriptomic sequencing under the condition of normal growth (14 days old, CK) and salt stress (150 mM of NaCl treatment for 14 days) to establish which genes responded differently in AS5 and NX420. The RNA sequencing of the 12 cDNA (CK420-1, CK420-2, CK420-3, S420-1, S420-2, S420-3, CKAS5-1, CKAS5-2, CKAS5-3, SAS5-1, SAS5-2, and SAS5-3) libraries was generated after filtering a total of 80.7 Gb clean bases; the average percentages for Q20 and Q30 were 96.2% and 87.7%, respectively (Table 3). Furthermore, using the HISAT2 software, 82.69% to 88.09% of the clean reads were mapped to the reference genome B73_V4. We investigated the number of upregulated and downregulated genes at the seedling stage and found that the number of upregulated genes was greater than that of the downregulated genes; there were 1854 genes upregulated and 1653 genes downregulated in AS5 (CKAS5 vs. SAS5) and 1826 genes upregulated and 934 genes downregulated in NX420 (CK420 vs. S420) (Figure 3A).

The KEGG enrichment results showed the DEGs from the treatment with and without 150 mM of NaCl for 14 days in AS5 and NX420. For NX420 and AS5, the top 20 KEGG pathways were based on the rich factor (Figure 3B,C). Common pathways such as metabolism pathways, the biosynthesis of secondary metabolites, starch and sucrose metabolism, and nitrogen metabolism were evidently enriched in both NX420 and AS5 (Figure 3B,C), while plant hormone signal transduction and the MAPK signaling pathway related to salt stress were explicitly found in NX420 (Figure 3B), and alanine, aspartate, and glutamate metabolism were explicitly only found in AS5 (Figure 3C).

A Gene Ontology (GO) analysis was performed to explore the biological process, the cellular component, and the molecular function related to salinity tolerance in NX420 and AS5 at the seedling stage. The top 20 GO terms were screened out based on enriched gene numbers (Figure 3D–I). Common GO terms such as protein phosphorylation were evidently enriched in AS5 and NX420, and oxidation–reduction processes related to salt stress were evidently enriched in the two inbred lines for biological processes (Figure 3D,G). The response to abscisic acid, the glutamate metabolic process, and the positive regulation of the response to salt stress processes were only enriched in NX420 (Figure 3G). Common GO terms such as the integral component of the membrane and the plasma membrane were evidently enriched in AS5 and NX420 for the cellular component (Figure 3E,H). Common GO terms such as protein serine/threonine kinase activity and cation binding were evidently enriched in AS5 and NX420 for the molecular function (Figure 3F,I). In addition, oxidoreductase activity and transmembrane transporter activity related to salinity tolerance were enriched in the two inbred lines (Figure 3F,I); ATP binding was only enriched in AS5 (Figure 3F).

### 2.5. The Candidate Genes’ Combined Analysis of DEGs and BSA-seq

The BSA-seq results show that 106 candidate regions for salt tolerance were mapped on all of the chromosomes by BSA-seq using the extreme populations. Based on the polymorphisms identified between parents, we detected 77 genes containing non-synonymous and synonymous coding SNPs and frameshift mutations in the open reading frame (ORF) regions. With transcriptome sequencing, we detected a large number of differentially expressed genes (DEGs) at the seedling stage under salt stress. The overlap of BSA-seq and DEGs showed that were two genes in AS5, *Zm00001d053925* and *Zm00001d037181*, and four genes in NX420, *Zm00001d053925*, *Zm00001d037181*, *Zm00001d0039920*, and *Zm00001d045404*. Among them, three genes were significantly enriched in the integral component of the membrane. In addition, *Zm00001d053925* and *Zm00001d037181* were both detected in AS5 and NX420 (Table 4).

### 2.6. Candidate Gene Confirmation Using Real-Time PCR

Two candidate genes, *Zm00001d037181* and *Zm00001d053925*, were finally determined through transcriptome and BSA techniques. To further confirm whether the candidate genes respond to salt stress and have different expressions in the two parents, 150 mM NaCl stress was applied to maize seedlings 10 days after sewing, and the leaves at 0 h, 24 h, and 48 h after treatment were taken for real-time PCR verification. The transcriptional levels of *Zm00001d053925* in the AS5 leaves at 24 h were 7.83 times those at 0 h, while in NX420, its expression was only 1.46 times that at 0 h. After 48 h of treatment, its expression in AS5 was 41.99 times that at 0 h, while in NX420, its expression was only 6.06 times that at 0 h (Figure 4A). In the root, the NaCl treatment decreased the mRNA levels in NX420, while it increased the mRNA levels in AS5 (Figure 4B). The PCR results showed that *Zm00001d037181* was expressed a little bit higher in NX420 than in AS5 leaves and roots at 24 h. There was no significant difference at 48 h (Figure 4C,D). These results indicate that *Zm00001d053925* responds to salt stress and has different expression patterns in different maize inbred lines.

*Zm00001d053925* is a protein with an unknown function and encodes 289 amino acids. GO analysis shows that it is an integral component of the membrane protein (Table 4). The protein structure prediction shows that there are five transmembrane domains (Figure 5A); the first is located at 56–73, and the second is located between 93 and 114. The third is between 192 and 214, the fourth is between 226 and 246, and the fifth is between 258 and 279. KEGG analysis showed that the gene was not enriched in any metabolic pathway.

## 3. Discussion

World, farmlands are increasingly affected by soil salinization, and most crops are glycophyte species that are sensitive to salt stress. As a result, salt tolerance is emerging as an important agronomical trait of crop breeding. However, decades of efforts have made little progress due to the complexity of salt responses and their interaction with variable environmental factors [2,28,29]. Maize is an important food crop worldwide, and it constitutes more than one-half of global calorie consumption [30]. However, salinity stress is a major threat to the development of maize production. Maize is sensitive to salt stress but is often planted on salt-contaminated land because most farmland is salinized. Therefore, elucidating the genetic architecture of salt tolerance in maize is instrumental for improving its salt tolerance. In this study, two maize inbred lines (AS5 and NX420) collected locally from Ningxia Agricultural Institute, Ningxia Province, China, were used due to their significant differences upon salt treatment (Figure 1A). BSA-seq and RNA-seq were combined to obtain SNPs and DEGs based on the plant height and the loss of biomass after growth in salt conditions. Similar strategies have been widely used in salt-tolerant maize. For example, compared with Zheng58 or Jing742, Chang7-2, LH65, and D9H showed salt sensitivity with a greater loss of biomass and dry leaves [31,32]. In our case, the loss of biomass in NX420 was around 40% higher than that in AS5 (Figure 1C), indicating that these two lines had more severely different phenotypes for salt tolerance analysis.

The analysis of Na^+^ and K^+^ also indicated that new candidates could be found using these two inbred lines because their main differences in Na^+^ and K^+^ were detected in both the leaves and roots, like in most cases, but salt decreased K^+^ in both the shoots and roots, resulting in no differences in the roots between these two lines [15,31]. Similar to other genes involved in salt tolerance, such as HAK4, HKT2, and SOS1 [15,16,31], the transcription levels of *Zm00001d053925* were induced upon salt treatment, especially in AS5, implying that its higher mRNA levels could be related to the higher K^+^:Na^+^ ratio in the leaves (Figure 1H and Figure 4). However, after salt stress, there was no significant difference in the content of Na^+^ and K^+^ in the roots of the two inbred lines, and we also found that there was no significant difference in the transcription level of *Zm00001d053925* in the two inbred lines (Figure 1E,G and Figure 4). We speculated that the gene might regulate the balance of sodium and potassium ions in the leaves. This new gene located on chromosome 4 was different from *ZmSOS1* (chromosome 1) [31], *ZmHKT1* (chromosome 3) [33], and *ZmHAK4*, is preferentially expressed in the root stele, and encodes a novel membrane-localized Na^+^-selective transporter that mediates shoot Na^+^ exclusion, probably by retrieving Na^+^ from xylem sap. Therefore, the function of the new gene may be different from that of ZmHAK4, which needs further functional analysis.

Maize salt tolerance is a complex trait which comprises distinct mechanisms, such as osmotic, ionic, and oxidative tolerance and other tolerances [1,34,35], making it challenging to identify previously uncharacterized QTLs that are related. Thus, in this study, we applied BSA and transcriptome techniques to identify a previously uncharacterized QTL which is related to maize salt tolerance. QTL mapping is a commonly used technology for dissecting complex trait loci in maize. For instance, QTL mapping has led to us understanding genetic regulation, including maize plant architecture [36], photoperiod sensitivity [37], flowering time [38], resistance to head smut [39], drought tolerance, and other traits [40,41]. Thus, in this study, we used BSA to detect the SNPs related to salt tolerance in the maize seedling stage (Figure 1 and Figure 2 and Table 1 and Table 2). We identified 106 SNPs which were distributed on 10 chromosomes of maize (Appendix A), 7 on chromosome 1, 5 on chromosome 2, 25 on chromosome 3, 12 on chromosome 4, 4 on chromosome 5, 10 on chromosome 6, 19 on chromosome 7, 13 on chromosome 8, 1 on chromosome 9, and 2 on chromosome 10. Previous studies on QTL in maize for salt tolerance have identified, in the seedling stage, ZmNC1 and ZmSTL2 located on chromosome 1 [31,42], ZmSTL1 located on chromosome 3 [33], ZmqKC3 located on chromosome 5 [16], and so on, and these QTL findings are different in our study for chromosomes 1, 3, and 5. Previous studies also found that ZmNC2 (Chr4-57153152) was located in an intergenic region on chromosome 4. The key candidate region detected in this study, Chr4-243690332, is not in the same region as ZmNC2, and the candidate gene Zm00001d053925 is in this region (Appendix A). Therefore, there may be several key QTLs on chromosome 4 that are related to salt tolerance in maize in the seedling stage.

Thus, based on the polymorphisms identified between both parents, we detected 77 genes associated with salt tolerance using BSA. In addition, the overlapping of BSA-seq and DEGs showed that there were two genes in AS5 and NX420: *Zm00001d053925* and *Zm00001d037181*. Among them, three genes were significantly enriched in the integral component of the membrane. In addition, *Zm00001d053925* and *Zm00001d037181* were both detected in AS5 and NX420 (Table 4); the transcriptional levels of *Zm00001d053925* in the AS5 leaves at 24 H were 7.83 times those at 0 H, while in NX420, its expression was only 1.46 times that at 0 H. After 48 H of treatment, its expression in AS5 was 41.99 times that at 0 H, while in NX420, its expression was only 6.06 times that at 0 H, and for *Zm00001d037181,* there were no obvious differences in the two inbred lines (Figure 4A). Therefore, *Zm00001d053925* as a candidate gene involved in salt tolerance has been identified in this study, and *Zm00001d053925* is a protein with an unknown function and encodes 289 amino acids. Additionally, the predicted protein structure contained five transmembrane domains, which resembled the structures of an ion transporter, but these ion transporters contain at least eight transmembrane domains (Figure 5B–D), such as the HKT protein [43]. These results indicate that *Zm00001d053925* could be a new candidate involved in salt tolerance. In addition, its transmembrane domains were similar to those of known ion transporters, such as ZmHAK4 (Figure 5B), ZmHKT1 (Figure 5C), and ZmHKT2 (Figure 5D), but they had a different number of domains, which suggests that it might be involved in ion transport.

## 4. Materials and Methods

### 4.1. Materials, Plant Growth, and Salt Treatment

The maize inbred NX420 and AS5 were collected from Ningxia Agricultural Institute, Ningxia Province, China. To measure the leaves and roots’ Na^+^ and K^+^ contents for NX420 and AS5, pots with a diameter of 10 cm and a height of 10 cm were filled with a uniformly mixed Pinstripe substrate and were watered to soil saturation with 150 mm of NaCl solution (Salt) or watered to soil saturation with water (Control). Ten seeds were planted in each pot and then grown in a glasshouse for 2 weeks; the leaves and roots were collected for the measurement of Na^+^ and K^+^ contents and to measure the height and biomass (fresh) of NX420 and AS5. The statistical analysis of the data used Graphpad prism9. 

### 4.2. Measurement of Ion (Na^+^ and K^+^) Content

The Na^+^ and K^+^ contents were determined with reference to the method described previously [44]. In essence, the leaves and roots were dried at 80 °C for 24 h to a constant weight, the dry weights of the samples were classified as the dry biomass, and the samples were then incinerated in a muffle furnace at 300 °C for 3 h and at 575 °C for 6 h. Next, 10 mL of 1% hydrochloric acid was used to dissolve the ashes, and then the samples were diluted in 1% hydrochloric acid. Na^+^ and K^+^ concentrations were analyzed using a 4100-MP AES device (Agilent, Santa Clara, CA, USA).

### 4.3. The Construction of the Extreme Population

We generated F1 seeds by crossing NX487 and AS5, and then we obtained the F2 population by selfing the F1 plants. The F2 plants were grown under 150 mM NaCl conditions for 14 days. Subsequently, we selected 30 salt-tolerant F2 plants showing a high height phenotype comparable with that of AS5 as a tolerant population, and 30 salt-sensitive F2 plants showed a tall height phenotype comparable with that of NX420 as a sensitive population.

### 4.4. Sample Collection, Extraction of Genomic DNA, and Construction of Segregating Pools

For the BSA-seq analysis, the young leaves of each line were sampled 14 days after being planted. The cetyltrimethylammonium bromide (CTAB) method was used to obtain high-quality genomic DNA [45]. DNA from 30 plants representing tolerance and sensitivity was equally mixed to form tolerance and sensitivity pools [25].

### 4.5. BSA-seq Analysis (Finished by GGI Tech)

BSA-seq was used to identify the genes regulating salt tolerance in the F_2_ population. We selected 30 plants that were extremely tolerant and sensitive to salinity to create an extreme population. Genomic DNA was extracted using a DNA extraction kit (GeneBETTER, Beijing, China). The DNA quality was checked using a microplate reader and agarose electrophoresis. Library preparation was performed according to the manufacturer’s protocol; it can be found in https://www.yuque.com/yangyulan-ayaeq/oupzan/cfq29z, 13 September 2022, Whole Genome Sequencing Library Preparation (DNBSEQ) BGI-NGS-JK-DNA-001 A0. Genomic re-sequencing was conducted to generate paired-end 100-base (PE100) reads using the DNBseq platform, which was conducted by BGI (Beijing, China). Clean reads were aligned to the reference genome sequences of the B73 genome (https://www.ncbi.nlm.nih.gov/assembly/GCF_000005005.2/ (accessed on 13 September 2022), Zea_mays.AGPv4.dna.toplevel.fa) using BWA software [46]. SNPs and small InDels were detected using GATK4 software [47]. We excluded SNP/InDel positions with multiple genotypes and a read depth < four from the two bulk sequences. The association analysis was conducted with the Euclidean distance (ED) and SNP/InDel index [48,49]. The overlapped regions based on the above two methods were considered candidate regions for salinity tolerance. 

### 4.6. Transcriptome Sequencing

Three replicated plants that were sampled and collected were the same as those in 4.1; the leaves of NX420 and AS5 were used for transcriptome sequencing. There were three independent RAN-seq libraries per treatment, namely, Control: AS5-1, -2, -3, and NX420-1, -2, -3, Salt treatment: AS5-1, -2, -3, and NX420-1, -2, -3. Total RNA samples were extracted using TRIzol reagent (Invitrogen) and then treated by RNAase-free DNAse I (NEB) in order to remove genomic DNA. mRNA libraries were created according to the standard protocols provided by BGI. mRNA quality, including the mRNA concentration and fragment size, was tested by using Qubit2.0 and Agilent 2100. mRNA was enriched using Dynabeads oligo (dT) (Invitrogen, Waltham, MA, USA) and a fragmentation buffer. Double-stranded cDNAs were produced using reverse transcriptase (Superscript II; Invitrogen) and random hexamer primers and further purified using AMPure XP beads. The purified double-stranded cDNA samples were enriched by PCR to construct the final cDNA libraries for sequencing using Hiseq 2500 (150 bp paired ends), provided by BGI (China). 

Clean reads were also aligned to the reference genome sequences of the B73 genome. Gene expression differences among the different sample points were detected using the EBSeq package (vl.10.1). A fold change ≥ 2 and a false discovery rate (FDR) < 0.01 were set to act as the standards for screening the DEGs. Functional classification of the DEGs, including the Gene Ontology and KEGG pathways, was analyzed using the GOseq R package (Release2.12) and KOBAS software (v2.0).

### 4.7. RNA Extraction and Real-Time PCR 

To analyze the effect of salt stress on the *Zm00001d037181* and *Zm00001d053925* transcription levels, 10-day-old NX420 and AS5 plants were treated with 150 mM NaCl solution; then, the samples (leaves and roots) were collected at 0 h, 24 h, and 48 h. Total RNA was extracted using an RNA pure plant kit (Gene Better, Beijing, China); 2.0 μg RNA was used to synthesize first-strand cDNA using M-MLV reverse transcriptase, and qRT-PCR analyses were conducted using TransStart Top Green qPCR SuperMix (TRANS, Beijing, China) on an ABI 7500 thermocycler (Applied Biosystems, Marsiling industrial, Singapore). The maize GAPDH gene was provided as an internal standard. The primer sequences (ZmGAPDH-F, ZmGAPDH-R, Zm181-RT-F, Zm181-RT-R, Zm925-RT-F, and Zm1925-RT-R) are listed in Appendix A. A 2^−ΔΔCt^-based calculation was used to quantify gene expression.

## 5. Conclusions 

In conclusion, our QTL approach obtained *Zm00001d053925*, a new candidate gene involved in salt tolerance, which was identified with the combination of BSA-seq and transcriptome analysis based on the different phenotypes of two maize inbred lines. It encoded an unknown function protein which was predicted to contain five transmembrane domains, and its transcription levels were related to salt responses. Thus, a novel gene target for the development of salt-tolerant maize was identified, which provides a new gene resource for salt-tolerant maize research.

## Figures and Tables

**Figure 1 plants-12-01331-f001:**
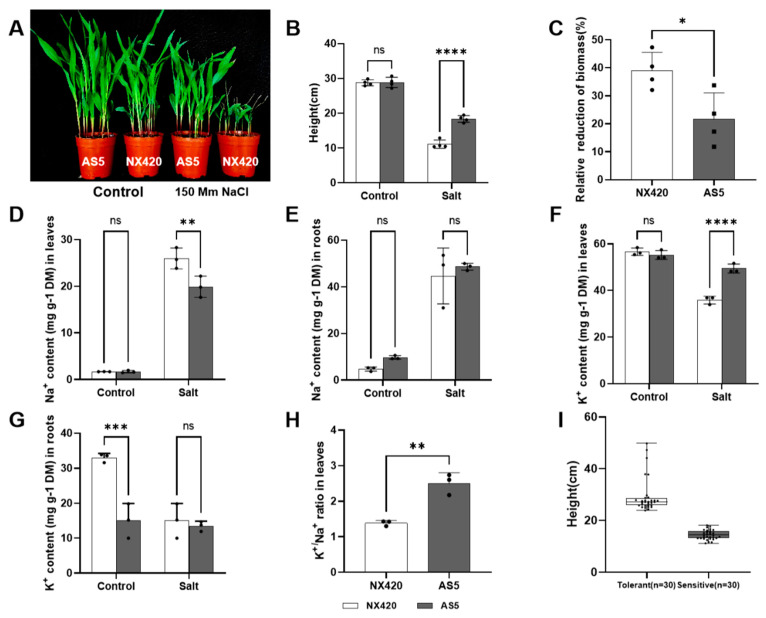
Identification of salt tolerance in AS5 and NX420. (**A**) The seedlings of AS5 and NX420 were treated with 150 mM NaCl for 14 days. (**B**,**C**) Plant height and the reduction in biomass. (**D**,**E**) Na^+^ content in leaves and roots. (**F**,**G**) K^+^ content in leaves and roots. (**H**) K^+^:Na^+^ ratios in leaves. (**I**) Plant height of extremum population. The bar is 10 cm. *, *p* < 0.1; **, *p* < 0.01; ***, *p* < 0.001; ****, *p* < 0.0001, ns is no significant difference.

**Figure 2 plants-12-01331-f002:**
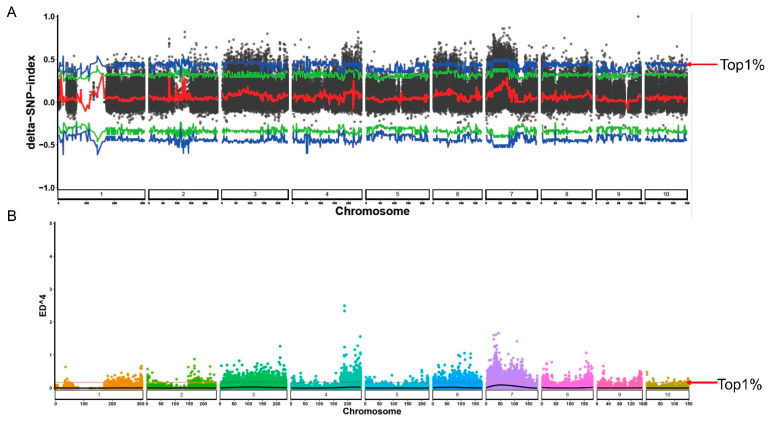
BSA: (**A**) SNP/InDel-index method and (**B**) ED method. Blue and red arrows indicate the top 1% threshold value in the two different methods, respectively.

**Figure 3 plants-12-01331-f003:**
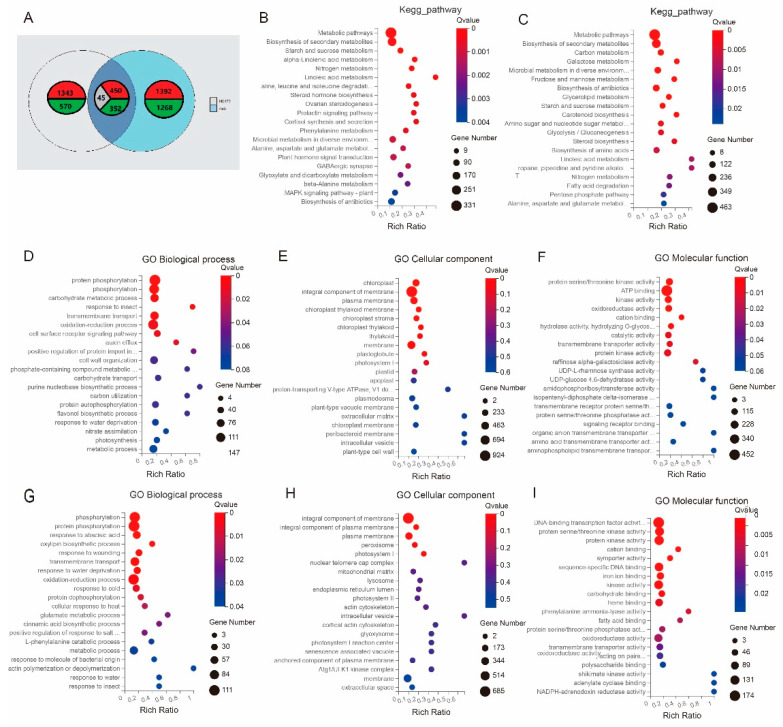
The differentially expressed genes (DEGs) at the seedling stage in NX420 and AS5. (**A**) The number of DEGs and two inbred lines; (**B**) KEGG enrichment at the seedling stage in NX420; (**C**) KEGG enrichment at the seedling stage in AS5; (**D**–**F**) GO enrichment at the seedling stage in AS5; (**G**–**I**) GO enrichment at the seedling stage in NX420.

**Figure 4 plants-12-01331-f004:**
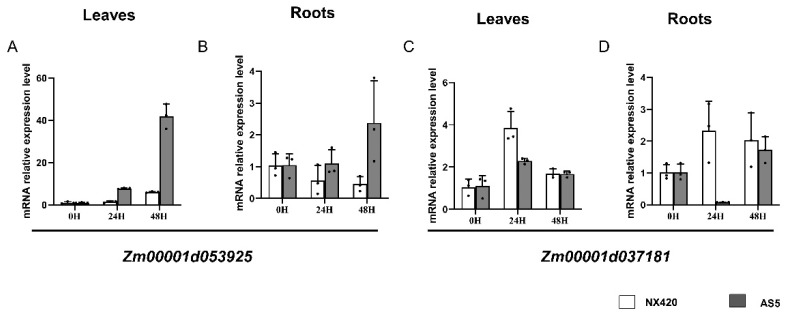
The mRNA relative expression level in the leaves and roots of NX420 and AS5. (**A**,**B**) The mRNA relative expression level of *Zm00001d053925* in the leaves and roots; (**C**,**D**) the mRNA relative expression level of *Zm00001d037181* in the leaves and roots.

**Figure 5 plants-12-01331-f005:**
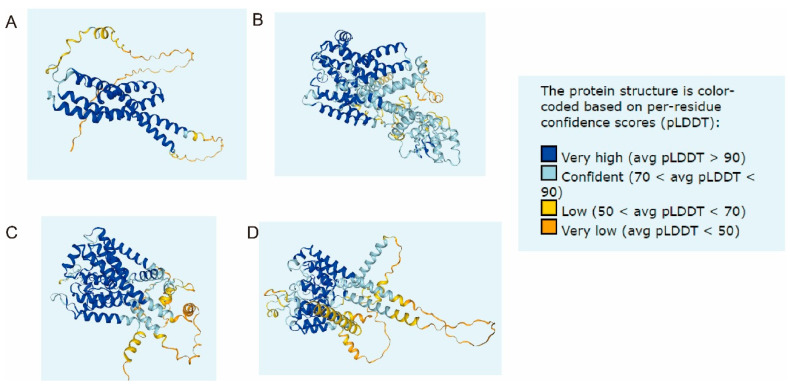
The prediction of the protein structure (www.maizegdb.org, 17 February 2022). (**A**) *Zm00001d053925*; (**B**) *Zm00001d049987* (ZmHAK4); (**C**) *Zm00001d040627* (ZmHKT1); (**D**) *Zm00001d014680* (ZmHKT2).

**Table 1 plants-12-01331-t001:** Coverage of the reads mapping to the B73 reference genome from the re-sequencing of AS5 and NX420 using BGI.

Sample Name	AS5 (Tolerant)	NX420 (Sensitive)	Tolerant	Sensitive
Mapping read	623,006,996	624,618,518	795,107,985	763,413,251
Mapping ratio	99.77%	99.81%	99.81%	99.81%
Clean read	616,749,238	620,812,254	788,519,740	757,382,400
Clean base	92,512,385,700	93,121,838,100	118,277,961,000	113,607,360,000
Depth	43.330X	43.615X	55.397X	53.210X
Properly paired read	574,440,938	591,572,262	743,200,496	591,572,262
Properly paired ratio	93.14%	95.29%	94.25%	94.41%

**Table 2 plants-12-01331-t002:** Distribution statistics of high-quality SNPs on chromosomes.

Chromosome	SNP Number
1	657,086
2	639,709
3	878,652
4	668,120
5	642,954
6	598,968
7	690,312
8	625,423
9	598,375
10	516,732
Total	6,516,331

**Table 3 plants-12-01331-t003:** Summary of mapping reads and RNA-seq.

Sample	Raw Reads (M)	Clean Reads (M)	Clean Bases (Gb)	Q20 (%)	Q30 (%)	Clean Reads Ratio (%)	Total Mapping (%)
CK420-1	47.33	43.99	6.6	96.37	87.99	92.95	88.93
CK420-2	49.08	45.26	6.79	96.34	88.25	92.23	87.76
CK420-3	49.08	45.4	6.81	96.47	88.49	92.51	87.57
S420-1	49.08	45.4	6.81	96.34	87.97	92.5	87.89
S420-2	50.83	45.06	6.76	96.39	88.2	88.65	85.84
S420-3	49.08	44.6	6.69	95.97	87.26	90.87	88.09
CKAS5-1	49.08	44.36	6.65	96.02	87.47	90.39	84.05
CKAS5-2	49.08	44.03	6.6	96.11	87.42	89.71	84.14
CKAS5-3	50.83	45.37	6.81	95.4	85.85	89.26	84.33
SAS5-1	50.83	45.1	6.77	96.53	88.53	88.73	82.69
SAS5-2	49.08	44.05	6.61	96.28	87.91	89.76	83.73
SAS5-3	49.08	45.36	6.8	96.18	87.75	92.42	84.28

**Table 4 plants-12-01331-t004:** The candidate genes’ overlapped BSA-seq and DEGs.

Sample	AS5		NX420			
ID	*Zm00001d053925*	*Zm00001d037181*	*Zm00001d053925*	*Zm00001d037181*	*Zm00001d039920*	*Zm00001d045404*
Chr	4	6	4	6	3	9
ED^4^	0.361	0.452	0.361	0.452	0.237	0.182
SNP-index	0.55	0.58	0.55	0.58	0.5	0.46
q-value	1.49 × 10^−7^	0.041122907	6.02 × 10^−6^	0.014798166	0.008950737	0.041086237
GO Annotation	Integral component of the membrane	-	Integral component of the membrane	-	Integral component of the membrane	Acid phosphatase activity

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
