# Peer review of "New Genes Identified as Modulating Salt Tolerance in Maize Seedlings Using the Combination of Transcriptome Analysis and BSA"

_plants, 2023, doi:10.3390/plants12061331_

Round 1

Reviewer 1 Report

The manuscript cannot be considered for publication due to following points:

(1) Screening and identification of genes for salt tolerance only upto 14 days old seedling will not hold much significance. Salt stress will have larger impact in the further physiological, reproductive and post-flowering phase of crop growth. Hence the results obtained in the study will not be much useful.

(2) Title: Use of tolerance is more appropriate for abiotic stress like salinity than the use of word resistance

(3) Line No. 15: Throughout the manuscript, 'tolerant' to be used while indicating the genotype and 'tolerance' to be used when describing the phenomenon

(4) Line No. 17 and 21-22: These two sentences are repetition of same.

(5) Line No. 24-25: Need to be improved for clarity

(6) Thorough editing of English language is required throughout the manuscript.

(7) Line No. 83: Bulked Segregant Analysis 

(8) Line No. 90-100: Results are presented as part of Introduction

(9) Discussion: The discussion presented in the manuscript is too brief and more elaborative discussion supporting the results is required to arrive at the conclusion.

(10) Conclusions: Needs significant improvement

Author Response

please the file

Reviewer 2 Report

Manuscript plants-2226353 – Reading report

Title “New genes identified to modulate the salt-resistance in maize seedling using the combination of transcriptome and BSA”

This manuscript describes a study to identify candidate genes for salt tolerance in maize, by combination of different approaches, phenotyping, NGS re-resequencing, BSA Pool-seq of extreme phenotypes, and transcriptomic. A strong point is that the study is complex and seems well designed and the results are abundant, relevant and are globally well presented. The abstract is complete and well written. However, the manuscript suffers of some major weaknesses, which reduce the value of the results for the scientific community. The weakest point are the discussions, which are poor and do not contribute to get full advantage of the results. Half of the Discussions (L244-253) recalls general information on salinity stress and the experimental strategy of the research. The real discussions are limited to L254-273. The second weak point is the section Materials and Methods, which lacks some fundamental information on experiment design, laboratory protocols, and data analysis. This point limits the possibility for the readers to understand the results. Part of the methods that are missing are described in the Results section but they should be gathered in the proper section. A third weak point is the English writing. There are several spelling or grammar errors, some sentences are incomplete or the punctuation and uppercase are misplaced. In some cases, it is not possible to unambiguously get the meaning of the text. The most critical points are detailed below. I cannot list all the minor errors and mistakes and the authors should have the manuscript checked by an English-native speaker or a professional language editing service.

Major comments

Introduction

Q1.

L95. Here the epithets “NX420” and “AS5” are cited for the first time. It is necessary to define what they are.

Results

Q2. L118.

This text lines describes the design of the bulks of tolerant and sensitive plants. This information with full details should be moved to Materials and Methods, where it is missing. Only a short description could be written here, to make the results clear.

Q3. L143-145.

It is not clearly explained whether the total number of 6.5*10^6 SNPs were only the polymorphisms between the two bulks or also included the polymorphisms within the bulks. In addition, because of poor Methods, it is not clear whether or not the phenotypic data were included in this statistical analysis.

Q4. L160-163.

Same point as Q2. This text describes methods and would be better placed in the Materials and Methods.

Q5. L167-170 and Figure 3A.

I suggest to realize a Venn diagram to show how many up- or down-regulated genes are shared by the two maize lines or line-specific.

Q6. Figure 4

The symbols the represent the statistical significance of relative expression are not clearly visible in this small plots.

Q7. L236-239.

This period would be better placed in the discussions.

Materials and Methods

Q8 4.1.

The description of control plants is missing. The measurement of plant height and biomass is not complete. The authors present the effect of salinity on plant biomass, but it is not clear how the biomass was measured as fresh or dry matter.

How was the salinity tolerance of F2 plants calculated in order to rank the and select the extreme phenotypes?

The statistical analysis of phenotypic data (plant traits and ion content) is not described.

Q9 L289.

Chemical analyses. Write more information about the instrument 4100-MP AES, type of instrument, manufacturer.

Q10. 4.3 and 4.4

Cite a reference for the CTAB DNA purification protocol (L 292) and DNA kit (L 301).

Q11 4.3 and 4.4.

Describe the protocol used for pooling DNA of the bulk samples.

Provide a full description of NGS methods (wet lab and bioinformatics analyses) for all NGS analyses (genome re-sequencing of inbred lines and Pool-seq of bulks, RNA-seq). In particular, describe library construction (kit or protocol), sequencing protocol and instrument, sequencing coverage and depth, trimming and filtering of raw NGS reads, alignment to reference genome, SNP calling and filtering. Some of this information is provided but some steps are not fully described.

Moreover, the two statistical methods applied to identify tolerance-associated SNPs, i.e. “Euclidean Distance” and “SNP/InDel-index” have to be fully described, with reference to literature sources. For all bioinformatics analyses, the software applied and the main configuration parameters have to be reported.

Q12. Integration of sequence data of inbred lines and bulk samples.

In Materials and Methods 4.3 it seems the the inbred parent lines were re-sequenced but it is not clear how these data were used in the study. Were the sequence data of parent lines included in the data analysis of BSA-seq?

Q13 4.5 Transcriptome sequencing.

Describe the experimental design, i.e. how many replicates plants, and many independent RNA-seq libraries per treatment?

Round 2

Reviewer 1 Report

The authors have addressed the comments/ suggestions.

Author Response

Thank you.

Reviewer 2 Report

Manuscript plants-2226353 –V2 Reading report

Title “New genes identified to modulate the salt-resistance in maize seedling using the combination of transcriptome and BSA”

Overall evaluation

This is the first revision of the manuscript. I recall here the main criticisms and how the authors have addressed each point.

First criticism. Poor discussions.

The discussions have been greatly improved, with interpretation of the results and comparison with previous studies. However, repeats and non-relevant text are still present in this section and should be deleted. Moreover, in major comments I suggest some interesting points that are worth of further discussions. Altogether, the discussions still require some revisions.

Second criticism. Materials and Methods missing fundamental information.

Almost all the materials and methods are now fully described, with the exception of statistical analysis of quantitative data, which is still missing.

Third criticism. English language.

The whole manuscript appears greatly improved in terms of readability and clarity. Some minor points still remain and are listed below.

The present version of the manuscript has been deeply revised and improved according the reviewer’s comment. Two main weaknesses still remain. The first point is the lack of description of statistical methods, which probably escaped to the authors’ attention. The second point is the discussion about the chromosome location of genomic regions and previous QTL regions. This part has been added in this revision. Both points are commented below.

Major comments

Q1.

Abstract. Add some results about the functional annotation of the new candidate genes.

Introduction

Q2

L48-59. This text about Sodium uptake and exclusion. is very detailed and not relevant to introduce this study. It should be deleted or shortened.

Q3

L84-96. This paragraph reports a detailed summary of the main results and appears as a repetition of the abstract. It should be deleted or summarized.

Discussions

Q4

L242-251. This paragraph is to be deleted as it is not relevant to discuss the results.

Q5

L266-275. An interesting point to discuss is the different ion balance (Na and K) and gene transcript level observed in roots when compared to leaves.

Q6

L276-L293. The authors compare the genomic regions identified by BSA to the QTL for salinity tolerance previously identified in maize. The authors simply compare the correspondence of chromosomes where QTLs were detected. This approach is not appropriate, as genomic regions and QTL from different studies can be on the same chromosome but at different positions and including different genes. For a correct analysis, the authors should compare the genomic regions of this study with the projections (genome coordinates) of QTL intervals on the reference genome in order to search overlaps.

Q7

L294-303. The text newly added is simply a detailed repeats of results without interpretation. It needs to be summarized.

Materials and Methods

Q8

This section still lacks a paragraph to describe the statistical analyses of quantitative data, as biomass, height, chemical composition, ion ratio, and relative transcript level. How the statistical significance of differences and ratio was determined? Describe the statistical tests and possibly the software used.

Q9

Paragraph 4.6. Transcriptome sequencing.

Describe better the experimental design, i.e. how many replicated plants were sampled, and how many independent RNA-seq libraries per treatment and control?

Minor revisions

L22. Replace “chromosome” with “chromosomes”

L25 and in the whole manuscript. Replace “transcriptomic sequencing” with “transcriptome sequencing”

L43. The meaning of the sentence is not clear. Do you mean “ … an effective way to improve the cultivation of saline and and alkaline land”?

L81. Correct as follows “ … to find new genes associated with several traits …”

L85. Cite here the parent line names, in this way “ …using two inbred lines, named “NX420” and “AS5”, which show differential …”

Fig. 1. Insert a plot colour legend, for grey and white bars, as in fig. 4

Fig. 2I. The last plot 2I is not described in the caption text.

Fig. 3. Enlarge the size of the new Venn Diagram or place it in a separate figure.

Fig. 4 The symbols that represent the statistical significance of relative expression are not clearly visible in this small plots. If statistical significance is presented, the description of p threshold needs to be included in the caption, as in fig. 1.

L202. “chromosome” change to “chromosomes”; “population” change to “populations”

L203. Delete “both”

L205 “transcriptome sequencing”

L279. Change as follows “ … to identify previously uncharacterized QTLs which are related …”

L321. Replace “ … and watered” with “ … or watered”.

L348. Delete “of the construction using NX420 and AS5”

L352. Replace “find” with “found”.

Author Response

please see the document.
